# Monitoring of Inflammatory Bowel Disease in Pregnancy: A Review of the Different Modalities

**DOI:** 10.3390/jcm12237343

**Published:** 2023-11-27

**Authors:** Reem Al-jabri, Panu Wetwittayakhlang, Peter L. Lakatos

**Affiliations:** 1Division of Gastroenterology and Hepatology, McGill University Health Center, Montreal, QC H3G 1Y6, Canada; reem.aljabri@mail.mcgill.ca; 2Gastroenterology and Hepatology Unit, Division of Internal Medicine, Faculty of Medicine, Prince of Songkla University, Songkhla 90110, Thailand; 3Department of Oncology and Medicine, Semmelweis University, 1085 Budapest, Hungary

**Keywords:** inflammatory bowel disease, pregnancy, small intestinal ultrasound, fecal calprotectin, CRP, colonoscopy, sigmoidoscopy, MRI, CT

## Abstract

Inflammatory Bowel Disease (IBD) significantly affects women in their reproductive years. Understanding the relationship between IBD and pregnancy is crucial, given its impact across pre-gestational, gestational, and postpartum phases. Monitoring IBD activity during pregnancy involves various modalities. This review discusses these modalities, focusing on the efficacy and safety of Small Intestine Ultrasound (IUS) as a noninvasive and reliable option. While IUS has gained popularity, its technique-sensitive nature necessitates trained staff for optimal usage.

## 1. Introduction

Inflammatory bowel disease (IBD) has a high prevalence in ages before 35 years and affects 0.35% to 0.7% of the Western population [1,2]. The importance of studying the relationship between IBD and pregnancy is based on the peak prevalence of this disease during the reproductive period. Women with IBD have many concerns about the possibility of being pregnant and the pregnancy complications that may related to IBD. This causes them to overestimate the risks and delay their conception, leading to voluntary childlessness, which presents in about 18% of IBD women compared to only 6% of women in the healthy population [3,4].

Multiple studies showed comparable pregnancy probabilities among individuals with IBD, encompassing ulcerative colitis (UC) and Crohn’s disease (CD), when disease control is optimal and surgical history is absent [5,6,7,8]. Of note, a history of abdominal surgical interventions, particularly ileal pouch–anal anastomosis (IPAA) surgery, significantly delays conception and increases the risk of infertility [9,10] due to increased adhesions and subsequent fallopian tube obstruction and scarring [11]. However, new laparoscopic approaches could avoid this risk [12,13,14].

In addition to abdominal surgery, disease activity significantly impacts infertility risk. The majority of patients with a quiescent disease course before pregnancy experience favorable pregnancy outcomes. Conversely, active disease or flare-ups during or before pregnancy elevates the risk of adverse outcomes, including fetal loss, low birth weight, and preterm birth [15]. Padhan et al. [16] found that the rates of cesarean sections and unwanted fetal outcomes were significantly higher after the disease’s onset compared to periods before the disease’s onset. Additionally, IBD severity increases the risk of preterm delivery [17]. Ban et al. [18] found that flaring of the disease increased the risk of infertility compared to patients in remission, possibly due to a decreased ovarian reserve and inadequate weight gain [19,20,21]. To conclude, the activity of the IBD serves as an essential indicator of unfavorable pregnancy outcomes such as low birth weight, preterm birth, and small for gestational age (SGA).

Assessing and monitoring the IBD activity before and during the pregnancy is thus crucial to avoid any adverse outcome. Preconception counseling has been found to improve pregnancy outcomes, decrease the risk of flares, and improve drug adherence [22]. Nguyen et al. [23] recommend a three-month steroid-free remission before conception.

The present review provides a comprehensive review of the current literature on the available modalities for assessing and monitoring IBD activity during pregnancy. We aimed to elucidate the advantages and disadvantages of each diagnostic and monitoring modality and to discuss its role in the routine clinical practice of IBD management.

## 2. Methods

This review was conducted through a comprehensive search of electronic databases, including PubMed, Web of Science, and EMBASE, focusing on publications from the last two decades up to June 2023. Key search terms included “inflammatory bowel disease”, “pregnancy”, “monitoring”, and “intestinal ultrasound”, among others. We selected studies that specifically addressed the monitoring modalities of IBD during pregnancy, with an emphasis on the use of Small Intestine Ultrasound (IUS). The inclusion criteria were studies published in English that provided data on the efficacy, safety, and outcomes of IUS in pregnant patients with IBD. Exclusion criteria were studies not focusing on pregnancy, case reports, and editorials. The information was synthesized to provide an overview of current practices and the role of IUS in the clinical management of IBD in pregnant women.

## 3. Patient Report Outcomes and Clinical Symptoms

Indices such as the Harvey–Bradshaw Index (HBI), partial Mayo score, and Simple Clinical Colitis Activity Index (SCCAI) are noninvasive IBD assessment indicators validated in the context of non-pregnant patients, which are often used as basic cutoffs in clinical trials to evaluate the efficiency of an ongoing therapy [24,25,26]. Pregnant patients, however, are not entitled to these trials, and their scores for these indices may not support the clinical findings, e.g., endoscopic appearance. Close monitoring of pregnant patients should be done through frequent visits, to avoid any unfavorable outcomes. Physicians are reluctant to perform an endoscopy on a pregnant patient, which is the proven gold standard for the detection of bowel inflammation in non-pregnant patients. Clinical scores of The Self-Regulation Assessment (SeRA) and fecal assays of pregnant patients also do not appear homogenous [27]. Pregnancy symptoms such as fatigue, abdominal cramps, and rectal hemorrhage are liable to be misinterpreted as an active IBD, which should be validated in larger clinical trials with healthy patients. Thus, pregnant patients need alternative diagnostic modalities that are uninfluenced by physiological pregnancy changes.

### 3.1. Blood/Fecal Biomarkers

Several biomarkers are used for assessing IBD activity, for example, CRP, albumin, erythrocyte sedimentation rate (ESR), and hemoglobin; however, consideration should be given to the altered levels of biomarkers during pregnancy [28,29,30]. Fecal calprotectin (FCP) is an important biomarker for pregnant patients, as it remains unaffected by the pregnancy [31]. C-reactive protein (CRP) may be relied upon more than the ESR for the detection of flares [28], while endoscopy remains a more invasive diagnostic tool for evaluating the disease activity [32].

### 3.2. Erythrocyte Sedimentation Rate (ESR)

ESR physiologically increases in the pregnancy two- to three-fold, the upper limit of normal values, specifically at the end of the first trimester because of the physiologically increasing values of fibrinogen and may normally increase to 40 mm per hour in the third trimester, which makes it unreliable to be used in monitoring disease activity during pregnancy [33].

### 3.3. C-Reactive Protein (CRP)

CRP is also an inflammation biomarker. When it comes to the detection of active IBD in a pregnancy, CPR is more accurate than ESR [28]. However, the first of the three trimesters of pregnancy can potentially have an impact [34]. According to a study by Kammerlander et al. [27], women with IBD activity during all gestation trimesters had substantially different CRP levels from women without IBD activity, while Julsgaard et al. [31] reported significantly higher CRP levels in patients with IBD activity within just the second trimester, without the first or the third. In a prospective study, Bal et al. [35] examined the relationship between high CRP and clinical disease activity in pregnant women with IBD. They reported greater median CRP numbers in patients with clinically active disease, as compared to inactive, at two points in their pregnancy: preconception (6.95 vs. 2.80 mg/L, *p* = 0.559) and the first trimester (24.75 vs. 6.00 mg/L, *p* = 1.000). However, in the second trimester (8.85 vs. 12.40 mg/L, *p* = 0.5923) and the third trimester (5.45 vs. 11.90 mg/L, *p* = 0.592), the median CRP was surprisingly found to be lower in the active IBD. Thus, while the study shows that CRP is a useful tool in monitoring early pregnancy for IBD activity, its results may be inaccurate in the later trimesters. This may be due to the fact that minor infections are common in later trimesters, which may have increased CRP in pregnancy patients without clinically active IBD. Further research is needed before making any association between IBD and the pregnancy state.

This controversy between the studies makes it difficult to rely on the results of CRP in detecting IBD activity in pregnant women [34].

### 3.4. Fecal Calprotectin (FCP)

FCP is considered a reliable biomarker, among others, in detecting IBD activity [36]. It is a calcium-binding protein that is responsible for innate immunity by its expression in the granulocytes, phagocytes, macrophages, and monocytes [37]. It is expressed during the inflammation of the gastrointestinal mucosa and rises in stool before other systemic inflammation biomarkers [38]. Also, it correlates with the findings of the endoscope in patients with IBD [39,40], and its sensitivity and specificity reach 85% and 75%, respectively, in diagnosing active patients with IBD [41,42], which makes it a reliable method for distinguishing between IBD and other non-inflammatory gastrointestinal diseases like irritable bowel syndrome [43].

A prospective study assessing the FCP concentrations in 46 pregnant women in five periods, including pre-pregnancy, postpartum, first, second, and third trimesters, found that pregnant women had an average FCP concentration of 131 µg/g (range 0–3600), as compared to 0 µg/g in controls (range 0–84) (*p* < 0.0001). A strong association of FCP was found with the physician’s global assessment (r > 0.80; *p* < 0.0001). Before (r = 0.66; *p* < 0.0001) and after pregnancy (r = 0.47; *p* < 0.003), FCP also showed a correlation with HBI/SCCAI, not during pregnancy, however. According to PGA, a cutoff concentration of FCP (250 µg/g) correlated with active IBD at all five periods of monitoring (*p* < 0.0002). No association was found between CRP and HBI/SCCAI (*p* > 0.05); however, there was a significant association between FCP (*p* = 0.0007) and PGA (*p* < 0.0002) in all five periods of time.

Similarly, another study by Kammerlander et al. [27] found that FCP level correlated well with a severe flare of IBD, and it was significantly higher in pregnant women with moderate to severe active IBD than those without active disease in all trimesters and even the preconception period, while this was not the case in mild disease flares.

In another study, 157 pregnant women were examined to find an association between the FCP levels during pregnancy and postpartum [44]. Significantly high levels of FCP were found in all patients with active IBD, correlating positively with PGA and other disease activity indices. It was also seen that patients who developed disease flare later during pregnancy also showed higher FCP levels compared to those who maintained remission [44]. Higher levels of FCP were also noted by Huang et al. [45] in women with active disease preconception and during pregnancy than those with inactive disease. Kammerlander et al. [27] observed a higher FCP level in 219 women during pregnancy with active moderate–severe disease (as evaluated by HBI/patients based SCCAI or GPA), being treated with anti-TNF, positively correlated to the biomarker levels during all the pregnancy phases. Moreover, at a cutoff value of 200 mg/kg, FCP sensitivity was found to be 69.7–80%, specificity 66.7–73.3% with a positive predictive value of 66.7–74.4% (preconception and during the three trimesters) with no clinical variations in the CRP, hemoglobin, or albumin values. It was found by Barré and co-workers that the fecal lactoferrin levels showed a significant association with the PGA, modified HBI for CD, and partial Mayo score for UC [46]. Julsgaard et al. [31], after conducting their study on pregnant health vs. pregnant IBD women (evaluated by HBI/SCCAI or PGA), concluded that the level of fecal calprotectin was higher in pregnant women than in the control group, claiming it to be an effective indicator for active IBD. Kanis et al. [47] observed that while FC was a good indicator of the active disease with an overall high specificity and sensitivity, it was not as effective in predicting relapse in the following phase of pregnancy.

### 3.5. Albumin and Hemoglobin

Albumin was mainly used to detect IBD activity, as the patients lose albumin and protein from their intestinal mucosa [36]. Also, anemia is one of the associated complications of IBD because of recurrent intestinal bleeding and poor iron absorption [48]. However, the accuracy of using albumin and hemoglobin in detecting IBD activity has not been deeply investigated in the literature. Kammerlander et al. [27] found no significant difference between mild, moderate–severe, or no IBD activity groups. Also, they found that their concentrations decreased over time from preconception to the third trimester because of the physiological hemi-dilutional status during pregnancy [49]. Hence, albumin and hemoglobin are poor predictors of IBD activity during pregnancy [34]; however, they can be used only as routine investigations to detect complications during pregnancy, as IBD pregnant women have increased risks of anemia and malnutrition compared to those without IBD [50,51].

## 4. Imaging during Pregnancy

Diagnostic methods such as ultrasound and magnetic resonance imaging (MRI) are the only two diagnostic modalities considered to be safe for pregnant patients. In contrast, ultrasound may be technically difficult during the third trimester. Imaging techniques can also be employed, but care should be taken to use these modalities as sparingly as possible to avoid radiation exposure [12]. While there have been concerns about fetal growth when using a computerized tomography (CT) scan, gadolinium-free magnetic resonance imaging (MRI) is known to be safe with pregnancies [52]. Intestinal ultrasound (IUS) is a popular new modality that, due to its safety and ease of use, has the potential to replace other imaging techniques as the preferred point-of-care diagnostic in pregnant patients with IBD [53].

## 5. Invasive Procedures

### 5.1. Endoscopy

Endoscopic procedures have generally limited indications in pregnancy even though they are reliable methods for detecting the disease’s activity [52]. Additionally, there are concerns related to sedation, such as the risk of aspiration, hypoxia, and teratogenic effects of drugs used in sedation or anesthesia [52]. In contrast to a colonoscopy, which needs anesthesia support if evaluation of the disease activity is needed, endoscopic procedures like non-sedated flexible sigmoidoscopy are least likely to cause complications for both the mother and the fetus [54]. Of note, the maternal position during these interventions may be challenging [36].

### 5.2. Colonoscopy

A systematic review including 100 lower gastrointestinal tract endoscopies of healthy pregnant women showed that only 6% of patients who underwent colonoscopy or sigmoidoscopy had side effects that were not significantly different among the three trimesters, which labeled colonoscopy as a low-risk procedure in pregnancy [33]. This was supported by de Lima et al. [22] who compared 42 pregnant IBD women who underwent lower endoscopic procedures, whether colonoscopy (in 13 patients) or sigmoidoscopies (in 33 patients), to IBD pregnant women who did not undergo endoscopic procedures, and they found that the pregnancy complications were minimal with two cases of spontaneous abortion in the group who underwent endoscopic procedures; however, this was statistically insignificant.

### 5.3. Flexible Sigmoidoscopy

Sigmoidoscopy is preferred over colonoscopy during pregnancy due to unnecessary preparatory procedures, the short procedure duration, and avoidance of sedation [55,56,57]. These findings encouraged the use of un-sedated flexible sigmoidoscopy as it was associated with a very low risk of maternal and fetal complications whenever it was used during the pregnancy trimesters [54]. This was also supported by the American Gastroenterology Association Pregnancy Care Pathway to use when necessary in any trimester [52]. Based on the findings by Ko et al. [54] that FCP was not a reliable marker of distinguishing between mild or moderate IBD flare, with around 15% of patients with high levels of FCP and negative endoscopic findings, Flexible sigmoidoscopy was encouraged to be used to assess the severity of the flare without relying solely on FCP value.

## 6. Noninvasive Imaging Procedures

### 6.1. Role of Radiology in Pregnant Patients with IBD

For routine monitoring of patients with IBD during pregnancy, the articles were conflicted about the safety of using imaging during pregnancy [12]. Some articles supported their use under specific conditions, while others opposed it [12,58]. Imaging that does not involve electromagnetic radiation is considered safe in pregnancy, e.g., MRI and IUS [59]. Moreover, the use of non-ionizing radiation throughout pregnancy is recommended rather than the use of ionizing radiation. The risks of ration to the fetus are variant in the form of; Intrauterine Growth Retardation (IUGR), microsomia, mental retardation, and organ malformation [36]. However, a systematic review and meta-analysis of pregnant women without IBD demonstrated that maternal exposure to a low dose of radiation had no significant relationship with low birth weight (<2500 g) of the infants (RR = 1.37, 95% CI 0.93–2.02) and stillbirth (RR = 1.19, 95% CI 0.79–1.77), while it had a significant associated with miscarriage (RR = 1.27, 95 CI% 1.13–1.440) [60].

### 6.2. X-ray and Pregnancy

The use of X-rays in pregnancy should be generally avoided. The maximum risk responsible for a 1 rad exposure is thousands of times less than the risks of malformations, abortion, or genetic disease [61]. One abdominal X-ray leads to fetal exposure of 0.1 rad [62]. Therefore, the use of X-rays is limited to emergency situations.

### 6.3. CT and Pregnancy

The use of CT is restricted in pregnancy because of the risk of elevated levels of ionizing radiation on the fetus and the mother [36]. However, these risks depend on the dose and time of exposure [63]. Therefore, it may be necessary in emergencies such as intra-abdominal sepsis if both MRI and IUS are not available after adjusting the setting to the lowest dose of radiation. An animal study showed that a dose higher than 100 mg/g of radiation can induce malformations [64]. Regarding the maternal risk, there is no risk affecting breast milk, while the only risk is for breast tissue, which is related to a CT chest [59].

### 6.4. Capsule Endoscopy in Pregnancy

Capsule endoscopy is contraindicated for pregnant patients, as per the guidance from the device manufacturers, due to a lack of research conducted within this group [65]. It is hypothesized that the enlarged uterus during pregnancy could slow the movement of the capsule through the digestive tract, potentially because of the uterus pressing on the intestines or the slowing effects of the hormone progestin. There have been only a handful of instances where capsule endoscopy was performed during pregnancy, such as one incident where it helped identify a bleeding jejunal carcinoid, which was subsequently surgically treated, with the patient eventually giving birth to a healthy child [66]. Despite these instances having positive outcomes for both mother and child, the evidence is too limited to establish official clinical guidelines. Capsule endoscopy remains an investigational procedure in the context of pregnancy. It should be reserved for situations where it is critically necessary, and the only other option may be gastrointestinal surgery.

### 6.5. MRI

In contrast to CT, MRI is a safe alternative for the assessment of IBD activity in pregnancy with low exposure to radiation [36,59]. Both the BSG and ECCO guidelines recommend MRI as a suitable imaging modality for IBD evaluation [67]. Besides its utility in obstetric care as a routine investigation, it can help assess and diagnose IBD, including CD in the terminal ileum during pregnancy [68]. There have not been any documented fetal adverse effects since the 1980s [63]. A large retrospective study from 2003 to 2015 that compared first-trimester MRI with a non-exposure MRI showed no significant elevation in the risk of different birth outcomes: stillbirth, death of the neonates, congenital anomalies, cancers, or hearing loss when the first trimester ends [69]. There is controversy about the safety of using gadolinium during pregnancy, which can pass through the placenta to reach the fetal circulation and amniotic fluid [69]. However, there is limited evidence on the impact of gadolinium in pregnancy. MRE can be done with almost similar accuracy without contrast [63]. However, MRI use is limited due to its availability and high cost compared with other investigations [70]. In CD, MRI can detect the activity, extent of small bowel disease, strictures, small bowel obstructions, fistulas formed intra-abdominally, and abscesses. There are variants of indications for using MRI over other radiological modalities in the IBD pregnant population when suspecting complications of the extra-luminal disease or progressive strictures in CD and when IUS is inadequate or unavailable [63]. Table 1 summarizes the pros and cons of monitoring modalities of IBD in pregnancy.

## 7. Utility of IUS in IBD

With the evolving treatment modalities, guidance from the monitoring strategies has become critical and more important. Since IUS does not require prior fasting status or any bowel preparations, this modality has been gaining popularity in clinical practice [53]. However, the optimal performance of IUS necessitates an expert operator and access to the latest ultrasound technology. A systematic review in 2011 emphasized that with well-trained operators, MRI, CT, and IUS exhibit equal efficiency [71]. Some facilities have used IUS as the modality of choice for IBD diagnosis, with increasing acceptance globally. ECCO has recently recognized a structured curriculum for the quality assurance of IUS performance in IBD management. IUS evaluation has been shown to be of prime importance in both UC and CD, exhibiting a positive correlation with endoscopic inflammation.

A recent systematic review and meta-analysis identified more than 21 US scoring system indices to assess the activity of IBD by ultrasound [72]. Few IUS scores are currently validated to either UC or CD with standardized endoscopic scores (Appendix A). The commonest parameters of IUS were Bowel Wall Thickness (BWT), color Doppler imaging, bowel wall stratification, loss of haustrations, and fatty wrapping [73].

### 7.1. Bowel Wall Thickness (BWT)

BWT is the most reliable measurement that can be consistently quantified to identify disease activity and severity of flare in pregnancy [74,75]. In a retrospective study, there was a significant correlation between the FCP and BWT in any of the affected segments (r = 0.26, *p* = 0.03) [76]. In a meta-analysis study, the cutoff value was adjusted to three mm of thickness, which showed the maximum sensitivity of the test. In comparison, a BWT of less than 3 mm predicts mucosal healing [77]. According to a systematic review published in 2019, the sensitivity of IUS in detecting patients with Crohn’s disease ranged from 75% to 94%, and the specificity ranged from 87% to 100%. They also showed the disease accuracy of IUS ranged from 53% to 89% and increased in acute phases of UC. In UC, the BWT ranges from five to seven mm, and can be thicker in severe cases [78,79,80,81]. Mucosa is the most common bowel layer affected, especially if the disease is active. For these patients, IUS is useful to determine and evaluate the disease extension as BWT, and the loss of haustrations, reflects the extension of the disease [82].

### 7.2. Bowel Wall Stratification

The intestinal wall involves multilayers and can be effectively assessed through IUS. The loss of stratification, either focally or extensive, revealed a high degree of inflamed intestinal segment, which can be examined using a high-resolution US probe [83]. A cross-sectional study showed that by using IUS, BWT > 3 mm had the highest sensitivity of 72% for detecting active disease as by endoscope (95% CI: 46–89) followed by loss of wall stratification at 61% (95% CI 36–82%), then fibrofatty proliferation and increased vascularization at 50% (95% CI 27–73) for each of them [83].

### 7.3. Hemodynamic Evaluation of Abdominal Vessels

IUS may also be used for the hemodynamic evaluation of abdominal vessels in UC. Patients without IBD demonstrated lower mean velocities in the outflow of the hepatic and portal veins, while higher values were found in patients with IBD. The resistant index was also analyzed from the superior and inferior mesenteric arteries, with lower values demonstrated among the IBD population compared to the control group [84]. However, hemodynamic assessment may require special preparations and may not be conclusive for diagnosis in mild disease activity cases.

### 7.4. IUS and Pregnancy

Pregnant women with IBD tend to experience complications during pregnancy, usually seen during active steroid-dependent disease at the time of conception or during pregnancy; however, a large population-based study also revealed a positive link between the adverse outcomes with the quiescent phase of the disease, and owing to these potential effects of treatment on the mother and fetus it is important to assess the status of the inflammation before initiating or changing the therapy in pregnant women with IBD [4]. Several assessing modalities like endoscopy and magnetic resonance imaging tend to be invasive and thus are not favorable during pregnancy. IUS, on the other hand, being a noninvasive modality not requiring radiation and having easy accessibility as a point-of-care modality, makes it an effective and optimal method to assess and monitor IBD during pregnancy through visualization of the five bowel walls and assessment of their thickness [77].

A prospective observational cohort study by Leung et al. [85] showed that IUS may detect subclinical inflammation in asymptomatic pregnant women with CD, and stratify CD activity in symptomatic patients. The study also highlights the utility of IUS over lab-tested biomarkers in monitoring pregnant patients with IBD, assessing the response to treatment, and avoiding unnecessary prescription of corticosteroids, immunosuppressants, or the potentiation of ongoing treatment therapy. IUS images are real-time and also provide information regarding extraintestinal pathologies such as abscess, reactive lymphadenopathy, or mesenteric hyper-echogenicity. However, cutoffs BWT distinguishing active from non-active disease have not yet been established with context to pregnancy. In addition, the studies demonstrating the correlation between IUS and FCP with IBD are weak and have not been able to draw a conclusion (Figure 1).

Similarly, a prospective cohort by De Vooged et al. [53] supported that IUS proved to be a noninvasive monitoring modality, showing undoubted results with accurate clinical correlation in all the trimesters of pregnancy and defined IUS accuracy during pregnancy in patients with IBD as 92% (95% CI 82–99.8%). However, the visibility of sigmoid and terminal ileum (TI) becomes less in the third trimester, with a sensitivity of 84% and a specificity of 98%. The accuracy of IUS is highest in the first to second trimesters of pregnancy since it is able to provide adequate images compared with that provided in the third trimester, due to the compression of the uterus on the bowels, including the TI. The view of TI with IUS was less feasible in the second trimester compared with that of the first trimester [53]. Additionally, a cohort study showed that in the third trimester, there was a decrease in the feasibility of IUS compared with that of the first trimester at the sigmoid colon (*p* = 0.023) as well [53]. An additional limitation of IUS was the patient’s BMI. IUS may be limited in patients with BMI > 28 kg/m^2^ in the first trimester and BMI > 31 kg/m^2^ in the second trimester [76]. IUS is considered to be more dependent on the patient’s BMI as compared to CT or MRI, while it is a potential modality for assessing the intra-abdominal complications related to IBD, having an accuracy of 90% or more for severe strictures and enteric fistulae, 87% sensitivity and 90% specificity [77]. Treatment algorithms may be developed for the management of pregnant patients using IUS-supported prognostic and monitoring, especially in areas where economic issues prevail.

## 8. Therapeutic Drug Monitoring in Pregnant Patients with IBD

Therapeutic drug monitoring (TDM) can be a critical component in managing pregnant patients with IBD [87]. During pregnancy, physiological changes can alter the pharmacokinetics of medications, making standard dosing potentially ineffective or unsafe [88]. Evidence suggests that TDM can guide adjustments in drug dosing to maintain therapeutic levels, mitigate flares, and minimize the risk of adverse effects. This is particularly relevant for IBD, where maintaining disease remission is crucial for maternal and fetal health [89]. For instance, anti-TNF agents, commonly used in IBD treatment, may require dosage adjustments to prevent subtherapeutic levels due to the increased volume of distribution in pregnant patients [90]. By optimizing drug concentrations through TDM, clinicians aim to ensure effective disease control while also reducing the potential for drug toxicity, thus supporting better pregnancy outcomes.

## 9. Remote Monitoring in Pregnant Patients with IBD

Telemedicine is increasingly recognized as an effective means to manage the complex needs of patients with IBD globally, offering financial benefits and improved health outcomes. Several studies reported that the use of telemedicine in monitoring patients with IBD resulted in increasing the quality of life, disease-specific knowledge, medical adherence, and patient satisfaction [91,92,93,94]. Moreover, telemedicine significantly reduced health-related costs and healthcare utilization [91,94,95]. In pregnant patients with IBD, Jogendran et al. [96] assessed the practicality of managing IBD during pregnancy through remote monitoring with a home FCP test (IBDoc) and a self-assessment tool (IBD Dashboard). The majority of participants (77%) consistently used the tools throughout pregnancy, with the end-of-study questionnaires revealing a preference for the IBDoc home test and an intention to use it in the future. The research highlighted a notable discrepancy of over 50% between clinical assessments and objective measurements of disease activity, suggesting the integration of clinical and objective markers could enhance disease activity monitoring. These findings demonstrated that remote monitoring of tight control management is feasible among pregnant people with IBD and can be adopted into current clinical practice and healthcare settings.

## 10. Treatment Response in Pregnant Patients with IBD

Several studies reported data regarding the accuracy of IUS in monitoring the treatment response in patients with IBD [97,98,99,100]; however, a very limited number of studies evaluated this in pregnant women. De Voogd et al. [53] showed that among 38 pregnant patients with IBD, IUS showed a high agreement with reference standards. With an overall accuracy of 92% for detecting IBD activity throughout pregnancy, IUS showed a sensitivity of 84% and a specificity of 98%. For CD and UC, the accuracy was 94% and 89%, respectively. Treatment decisions were influenced by IUS results, with active disease identified on IUS prompting treatment escalation in 13 instances. Despite its effectiveness, in two UC cases with active disease, IUS indicated quiescent disease, leading to treatment adjustment based on increased symptoms and fecal calprotectin levels. Another study by Flanagan et al. [76] demonstrated that IUS had high specificity (83%) and sensitivity (74%) in diagnosing active disease during pregnancy when compared to FCP levels over 100 µg/g. IUS showed that inactive disease scans had significantly lower median FCP levels than those indicating disease activity. Active disease on IUS often led to changes in therapy during pregnancy in 61% of cases with active disease versus 2% with inactive disease. These findings highlight that IUS accurately detected patients responding to treatment initiated earlier in pregnancy. However, further large-scale studies are required to validate these findings.

## 11. Conclusions

During pregnancy, ensuring the safety of the investigations is paramount when monitoring patients with IBD, to mitigate risks for both the mother and fetus. This review highlights the strengths of FCP and IUS as pivotal, noninvasive tools for monitoring disease activity in pregnant patients with IBD. FC offers a quantitative biomarker for inflammation, while IUS provides a visual assessment of intestinal changes, each with its own merits in guiding therapeutic decisions. However, their limitations are evident; FCP may not always correlate with the clinical symptoms, and IUS requires specialized skills for interpretation and can be influenced by patient anatomy during pregnancy. Despite these challenges, the combined use of FCP and IUS can significantly enhance the accuracy of IBD monitoring tailoring treatment to ensure optimal maternal and fetal health outcomes. Further studies are required to evaluate the feasibility and accuracy of this combined approach in improving the monitoring of IBD in pregnant women.

## Figures and Tables

**Figure 1 jcm-12-07343-f001:**
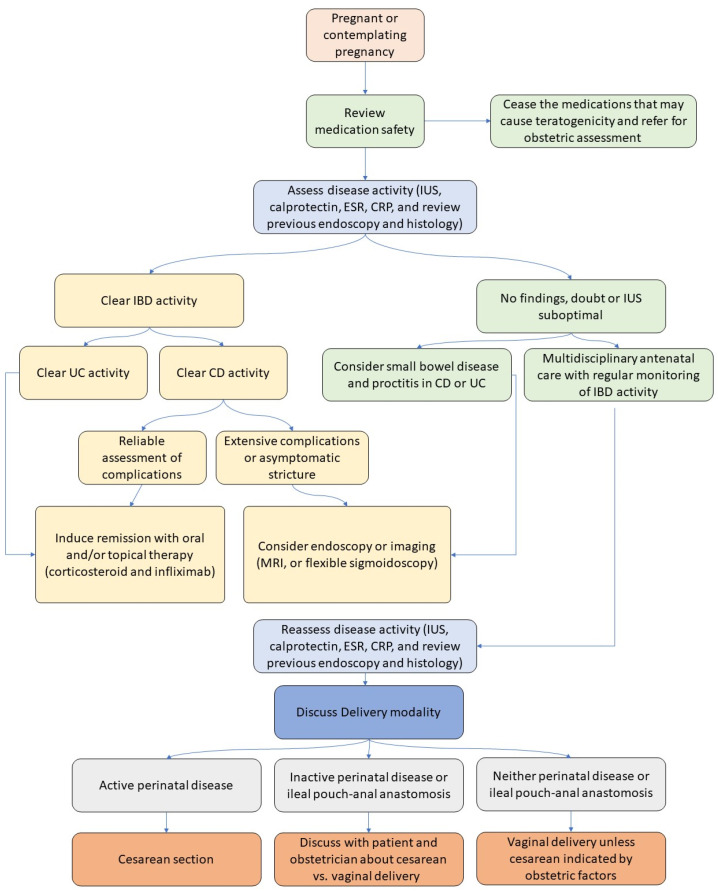
Point-of-care algorithm [86].

**Table 1 jcm-12-07343-t001:** Pros and Cons of monitoring modalities.

Monitoring Modality	Pros	Cons
Lower endoscopy		
Colonoscopy	The gold standard of disease monitoring. Early studies show no difference in adverse events between pregnant patients with IBD who underwent colonoscopy and those who did not undergo colonoscopy.	Limited studies Provider/patient hesitancy due to procedural and anesthetic concerns
Flexible sigmoidoscopy	It can be performed without sedation. No case reports of any procedure-related complications	Limited studies
Radiologic studies		
IUS	The safest form of radiologic imaging	Sensitivity in pregnancy is not well-established
	Contrast-enhanced ultrasound has been shown to have good results in IBD.	
Magnetic resonance imaging	No use of damaging ionizing radiation	No well-controlled studies of the teratogenic effects of gadolinium contrast in pregnant women have been performed, and the fetal risk is unknown.
	Can detect luminal and extraluminal abnormalities	
	Long-term safety after exposure to MRI during the trimester of pregnancy showed no increased risk of harm to the fetus or in early childhood.	
Biomarkers		
Albumin	Low albumin has been shown to be a predictor of poor outcomes in IBD	Limited utility in pregnancy due to pregnancy-induced hemodilution resulting in lower albumin values
ESR	Generally, it is a good marker of inflammation and reflects disease activity.	Limited utility in pregnancy due to physiologic increase in ESR (2–3× upper limit of normal)
CRP	Levels are only slightly raised in normal pregnancy and are still under the normal limits. CRP is higher in clinically active pregnant patients with IBD at preconception and first trimester compared to clinically inactive pregnant patients with IBD	It may not accurately reflect disease activity in the second and third trimesters. Limited studies in pregnant IBD population.
FCP	Measure of GI mucosal inflammatory activity detected before signs of systemic inflammation. Multiple studies show a correlation between FCP levels and noninvasive disease activity scores in CD and UC.	Conflicting evidence for the utility of FCP in IBD during pregnancy. Limited studies with actual endoscopic data to evaluate clinical activity

IUS, Intestinal Ultrasound; ESR, Erythrocyte Sedimentation Rate; CRP, C-react protein; FCP, fecal calprotectin; UC, Ulcerative colitis; CD, Crohn’s’ disease; IBD, Inflammatory bowel disease.

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
