# Peer review of "Monitoring of Inflammatory Bowel Disease in Pregnancy: A Review of the Different Modalities"

_jcm, 2023, doi:10.3390/jcm12237343_

Round 1

Reviewer 1 Report

Comments and Suggestions for Authors

This paper is timely and informative, addressing a relevant topic in the literature. Pregnant patients, especially those exhibiting features associated with a poor disease course, should receive regular monitoring. In this context, intestinal ultrasound (IUS), preferably in combination with fecal calprotectin (FCP), could play a pivotal role in the 'point-of-care' setting.

Please find below some points that need to be addressed:

Clarify the criteria you used for your review and whether you conducted a systematic search. Please provide details about the search criteria.

Please include evidence regarding the accuracy of IUS in detecting treatment responses in pregnant IBD patients.

As part of the monitoring strategy, therapeutic drug monitoring during pregnancy may be useful. Please provide evidence and include a comment on this.

Please include a paragraph on the role of capsule endoscopy during pregnancy

Recent studies have demonstrated the utility of remote monitoring in current clinical practice and the IBD healthcare setting. Please provide a comment on the feasibility of remote monitoring for tight control management in pregnant individuals with IBD.

I suggest including a table summarizing all the studies you mentioned on the use of FCP during pregnancy.

Furthermore, it may be valuable to provide a point-of-care algorithm for pregnancy

Conclusions should be more concise and focus on strengths and limitations of FC and IUS

Update the references to reflect the most current literature.

Comments on the Quality of English Language

The quality of english is good.  Minor editing are required

Author Response

Reviewer #1:

Comments and Suggestions for Authors

This paper is timely and informative, addressing a relevant topic in the literature. Pregnant patients, especially those exhibiting features associated with a poor disease course, should receive regular monitoring. In this context, intestinal ultrasound (IUS), preferably in combination with fecal calprotectin (FCP), could play a pivotal role in the 'point-of-care' setting.

Please find below some points that need to be addressed:

  1. Clarify the criteria you used for your review and whether you conducted a systematic search. Please provide details about the search criteria.

Thank you for your comment. This is not a systematic review, it is a narrative review; however, we conducted a systematic search, as detailed in the methods section.

  1. Please include evidence regarding the accuracy of IUS in detecting treatment responses in pregnant IBD patients.

Thank you for your suggestion, we have added a new section that discusses this point.

  1. As part of the monitoring strategy, therapeutic drug monitoring during pregnancy may be useful. Please provide evidence and include a comment on this.

Thank you for your comment, we do not think that this is quite relevant here; however, we added a brief comment about this point in our review.

  1. Please include a paragraph on the role of capsule endoscopy during pregnancy

Thank you for your comment, we added a paragraph about capsule endoscopy in pregnancy.

  1. Recent studies have demonstrated the utility of remote monitoring in current clinical practice and the IBD healthcare setting. Please provide a comment on the feasibility of remote monitoring for tight control management in pregnant individuals with IBD.

Thank you for your comment, we added a paragraph about remote monitoring in pregnant IBD patients.

  1. I suggest including a table summarizing all the studies you mentioned on the use of FCP during pregnancy.

Thank you for your comment, we have added a new table summarizing the two studies that discussed the role of FCP in IBD pregnant women.

  1. Furthermore, it may be valuable to provide a point-of-care algorithm for pregnancy

Thank you for your comment, we have added an algorithm based on your request.

  1. Conclusions should be more concise and focus on strengths and limitations of FC and IUS

Thank you for your comment. The conclusion section has been rewritten and improved based on your suggestion.

  1. Update the references to reflect the most current literature.

Thank you for your comment. We confirm that these references are pivotal, and we did our best to cite the most updated references.

  1. The quality of english is good.  Minor editing are required

Thank you for your comment, the entire manuscript has been revised.

Reviewer 2 Report

Comments and Suggestions for Authors

The manuscript is well written and displays information in targeted and well designed fashion. In my opinion, it lacks an illustration depicting both serum/feces biomarkers and so called invasive procedures. Also, authors named topic 2 as: Noinvasive biomarkers and 3 as: invasive. However, to assess the levels of CRP, albumin, ESR and hemoglobin its necessary to collect peripheral blood from individuals enrolled in the studies. Thus, I strongly suggest authors to rename those topics as follows: Topic 2 - Blood/Fecal biomarkers and Topic 3 - Invasive procedures, and Topic 4 - non-invasive imaging procedures.

Author Response

Reviewer #2:

  1. The manuscript is well written and displays information in targeted and well designed fashion. In my opinion, it lacks an illustration depicting both serum/feces biomarkers and so called invasive procedures.

Thank you for your suggestion. We have added a figure that shows the biomarkers of IBD. Figure 1 was created using BioRender.com

  1. Also, authors named topic 2 as: Noinvasive biomarkers and 3 as: invasive. However, to assess the levels of CRP, albumin, ESR and hemoglobin its necessary to collect peripheral blood from individuals enrolled in the studies. Thus, I strongly suggest authors to rename those topics as follows: Topic 2 - Blood/Fecal biomarkers and Topic 3 - Invasive procedures, and Topic 4 - non-invasive imaging procedures.

Thank you for your comment. We have changed it based on your suggestion.

Reviewer 3 Report

Comments and Suggestions for Authors

The article "Monitoring of inflammatory bowel disease in pregnancy: A review of the different modalities" is an interesting review article which has covered most of the available information in the field. Certainly, it is well verse that IBD has potential significant risk during pregnancy. The article explains in details the available diagnostic approaches with their limitations. Authors need to be appreciated for the efforts to put all this information in a single article in a very reader friendly manner.

Although this is a well-studied and well-written manuscript few minor suggestions for improving the manuscript are as follows:

1.    Few typos need to be corrected.

2.    Table 1 can be improved/formatted in better way.

3.    A table summarizing the methods with their practicality/limitations would be a noteworthy addition in the manuscript (from readers prospective).

4.       Later part of conclusion can be incorporated in discussion. Conclusion lacks a solid punch at the end with clear take home message or recommendation.

Author Response

Reviewer #3:

The article "Monitoring of inflammatory bowel disease in pregnancy: A review of the different modalities" is an interesting review article which has covered most of the available information in the field. Certainly, it is well verse that IBD has potential significant risk during pregnancy. The article explains in details the available diagnostic approaches with their limitations. Authors need to be appreciated for the efforts to put all this information in a single article in a very reader friendly manner.

Although this is a well-studied and well-written manuscript few minor suggestions for improving the manuscript are as follows:

  1. Few typos need to be corrected.

Thank you for your comment. The entire manuscript has been revised.

  1. Table 1 can be improved/formatted in better way.

Thank you for your comment. We have reformatted this table.

  1. A table summarizing the methods with their practicality/limitations would be a noteworthy addition in the manuscript (from readers prospective).

Thank you for your comment, we have added this table based on your suggestions.

  1. Later part of conclusion can be incorporated in discussion. Conclusion lacks a solid punch at the end with clear take home message or recommendation.

Thank you for your comment. The conclusion section has been rewritten and improved based on your suggestion.